# Preliminary Assessment of Burn Depth by Paper-Based ELISA for the Detection of Angiogenin in Burn Blister Fluid—A Proof of Concept

**DOI:** 10.3390/diagnostics10030127

**Published:** 2020-02-27

**Authors:** Shin-Chen Pan, Yao-Hung Tsai, Chin-Chuan Chuang, Chao-Min Cheng

**Affiliations:** 1Department of Surgery, Section of Plastic and Reconstructive Surgery, National Cheng Kung University Hospital, College of Medicine, National Cheng Kung University, Tainan 704, Taiwan; pansc@mail.ncku.edu.tw; 2International Center for Wound Repair and Regeneration, National Cheng Kung University, Tainan 704, Taiwan; 3Institute of Biomedical Engineering and Frontier Research Center on Fundamental and Applied Sciences of Matters, National Tsing Hua University, No. 101, Sec. 2, Kuang-Fu Rd., East Dist., Hsinchu 300, Taiwan; michaeltsai45@gmail.com (Y.-H.T.); jason20011122@gmail.com (C.-C.C.)

**Keywords:** partial-thickness burn injury, burn blister fluid, P-ELISA, angiogenin, burn wound healing

## Abstract

Rapid assessment of burn depth is important for burn wound management. Superficial partial-thickness burn (SPTB) wounds heal without scars, but deep partial-thickness burn (DPTB) wounds require a longer healing time and have a higher risk of scar formation. We previously found that DPTB blister fluid displayed a higher angiogenin level than SPTB blister fluid by conventional ELISA. In this study, we developed a paper-based ELISA (P-ELISA) technique for rapid assessment of angiogenin concentration in burn blister fluid. We collected six samples of SPTB blister fluid, six samples of DPTB blister fluid, and seven normal healthy serum samples for analysis. We again chose ELISA to measure and compare angiogenin levels across all of our samples, but we developed a P-ELISA tool and compared sample results from that tool to the results from conventional ELISA. As with conventional ELISA, DPTB blister fluid displayed higher angiogenin levels than SPTB in P-ELISA. Furthermore, our P-ELISA results showed a moderate correlation with conventional ELISA results. This new diagnostic technique facilitates rapid and convenient assessment of burn depth by evaluating a key molecule in burn blister fluid. It presents a novel and easy-to-learn approach that may be suitable for clinically determining burn depth with diagnostic precision.

## 1. Introduction

Burn wound prognosis depends on early and rapid diagnosis of burn depth. Superficial partial-thickness burn (SPTB) wounds heal spontaneously within two weeks of injury without scar formation. However, deep partial-thickness burn (DPTB) wounds take more than two weeks to heal and often result in hypertrophic scar formation if no aggressively surgical management. Optimal management of DPTB wounds can prevent skin scarring. Several methods were reported to assess the wound depth, including biopsy, thermography, laser Doppler techniques, and bedside clinical judgment [1]. Clinical observations are the gold standard for estimating clinical outcomes [2]. However, clinical assessment and prognosis of second-degree burn wounds with intact blisters become difficult in some cases, even for experienced surgeons. Measurements of tissue perfusion in injured wounds appear to be an option to assess tissue damage extent [3]. Although laser Doppler perfusion imaging was reported to be an efficient tool to evaluate blood flow of burn wounds [4,5,6], this machine is not readily available in all clinical settings.

Burn blisters, common to both SPTB and DPTB, are formed by an inflammatory response in early burn injury and exist between the epidermis and dermis [7]. The cytokines found in burn blister fluid can also be generated by activated or injured parenchymal cells after appropriate stimulation [8]. In our previous study, we observed that SPTB and DPTB blister fluids expressed different levels of angiogenin. When compared with SPTB, DPTB blister fluids displayed a higher angiogenin level [9]. In addition, angiogenin promoted in vitro angiogenesis such as endothelial cell proliferation and differentiation of circulating angiogenic cells as well as in vivo neovascularization [9]. Angiogenin, originally identified from a conditioned culture medium of colon cancer cells [10], is a potent angiogenic inducer and an independent prognostic factor in many cancers [11]. Because the angiogenin levels of SPTB and DPTB blister fluids were significantly different, the measurement of angiogenin in burn blister fluids can be used as a novel, noninvasive, or minimally invasive tool for surveying burn wound status.

ELISA is a well-established diagnostic tool for evaluating disease activity and assessing burn depth via analysis of fluid contents [9,12]. However, commercial ELISA kits are expensive, time-consuming, and are only available in clinical settings equipped with an ELISA reader. In order to simplify and expand clinical use of ELISA for early and efficient assessment of burn wounds, the development of a low-cost and rapid diagnostic tool for assessing burn depth is vital. Dot immunoassays on nitrocellular and filter papers is an established diagnostic tool [13,14,15]. Paper-based ELISA (P-ELISA), first developed by the Whitesides Research Group [16], has been an established diagnostic tool for decades. It is a useful procedure for performing immunoassays using a piece of filter paper for antibody–antigen recognition and has been applied for the diagnosis of infectious diseases (e.g., HIV and dengue fever), ophthalmological diseases (e.g., proliferative diabetic retinopathy, age-related macular degeneration), female genital diseases, and bullous pemphigoid [17,18,19,20]. The advantages of P-ELISA include speed, cost, small sample demands, and similar levels of sensitivity and specificity compared to conventional plate ELISA. Due to differential angiogenin levels between SPTB and DPTB blister fluids, the objective of this study is to develop a new application of the P-ELISA tool for the measurement of angiogenin expression in burn blister fluids in order to diagnose burn depth and facilitate improved burn wound care. Through conducting a small amount of clinical testing, we prove the clinical applicability of the device and guide the engineering process for further point-of-care and care-at-home devices with potential pre/clinical applications [21,22].

## 2. Material and Methods

### 2.1. Patient Samples

In order to study the differential expression of angiogenin in burn blisters from wounds of different depths, burn blister fluids were blindly aspirated with a needle from individually intact blisters within the first 3 days following injury and before identification of burn depth. Blister fluids were not classified into superficial or deep groups when fluids were harvested. Burn depth was confirmed by retrospective review of patient data according to healing status after the wound was healed. SPTB wounds were defined as those that healed within 14 days, and DPTB wounds were defined as those that required more than 14 days to heal or required burn wound debridement in the early assessment of wounds. Exclusion criteria included any potential bias such as fever, wound infection, or other severe medical problems including end-stage renal disease. Blood serum samples taken from healthy subjects were regarded as control samples. Informed consent was obtained from all patients, and study procedures were conducted in accordance with the Declaration of Helsinki and were approved by the Ethics Committee of National Cheng Kung University Hospital (No. A-ER-106-239, approval date 11 November 2017). All experiments were performed in accordance with relevant guidelines and regulations.

### 2.2. Preparation of Paper-Based 96-Well Plates Via Wax Printing

We designed a wax printing method for patterning Whatman No.1 filter papers (GE Healthcare, Buckinghamshire, UK) [20]. Briefly, a 96-well template was designed on a computer with Microsoft Office and then printed onto paper with wax. Complete, paper-depth hydrophobic barriers were created by melting the paper with printed-on wax at 105 °C for 5 min. Paper is highly permeable, which allows the wax to penetrate into the fiber matrix and then solidify to form defined hydrophobic barriers. This process is well-established and easy to complete.

### 2.3. Procedure of Paper-Based ELISA and Plate ELISA

Test zones were initially rinsed with 2 μL of Tris-buffered saline (TBS). Three microliters of burn fluid sample was then loaded onto the test zone and allowed to rest for 10 min. The test zone was then blocked with 3 μL 1% bovine serum albumin (BSA) for 10 min before adding 3 μL of rabbit anti-human angiogenin polyclonal primary antibody (Cat. No.: ab95389. Abcam, Cambridge, UK) and waiting another 10 min. Test zones were subsequently washed with 5 μL TBST (TBS+0.05% Tween 20), and then washed again with 10 μL TBST for a total of 2 washes. The washing process relied on a piece of blotting paper at the bottom to remove the washing buffer with capillary force. After washing, the test zone was incubated for 10 min with anti-rabbit IgG secondary antibody conjugated with horseradish peroxidase (HRP, Cat. No.: ab6721. Abcam, Cambridge, UK) before being washed again with 5 μL and 10 μL TBST (two washes). The color of the tested paper was developed after incubation with 3 μL of substrate solution (3,3’,5,5’-Tetramethylbenzidine (TMB): H_2_O_2_ = 1:1, Cat. No.: 555214. BD, Franklin Lakes, New Jersey, USA) for 10 min. The image signal was recorded using a camera at two time periods: (1) after 10 min of incubation with a secondary antibody; and (2) after 10 min of reaction time with the substrate solution. Both images were processed with Photocap software and analyzed with ImageJ software. To determine color intensity change, before and after images were first converted into 8-bit grayscale. Mean intensity was determined by comparing grayscale value differences from before and after images. Further normalization of the results was achieved by expressing values in terms of relative intensity, which was defined as (mean intensity of [experiment group]–mean intensity of [control group])/mean intensity of [control group]). Plate ELISA (Cat. No.: ab10600. Abcam, Cambridge, UK; Avastin^®^ (bevacizumab), Roche, Basel, Switzerland) was used to measure the angiogenin and VEGF (Vascular endothelial growth factor) values in burn blister fluids. Each sample was applied in triplicate, and the average data from two plates were taken as final readouts. Values were expressed as mean ± SD. To also examine the role of pH on burn wound status, burn blister fluid pH values were collected with pH test strips (Cat. No.: 92111. Machery-Nagel, Düren, DE).

### 2.4. Data Analysis

Differences in angiogenin levels between SPTB and DPTB blister fluids was assessed using the Mann–Whitney U test. Analysis of variance was used to determine statistical differences among multiple groups. The relationship between the titer from the conventional plate ELISA and the relative intensity of P-ELISA was correlated using Pearson’s correlation coefficient. Values of *p* < 0.05 were considered statistically significant. The results were expressed as mean ± SD.

## 3. Results

We have attempted to analyze the angiogenin levels in burn blister fluids to assess burn depth using P-ELISA. To determine optimal primary antibody concentration, we examined colorimetric responses to different dosages of recombinant angiogenin antibody on test paper impregnated with blister fluid. The results showed that 1 µg /mL of anti-angiogenin antibody provided the best signal. (Appendix A). To achieve the best staining with minimal background interference, we tested three different concentrations (0.05, 0.04, and 0.033 µg/mL) of secondary antibody. After testing serial dilutions of secondary antibody, we found that 0.033 µg/mL demonstrated the highest signal-to-background ratio for our primary antibody (Appendix A). We washed the test paper twice, first with 5 µL of TBST solution and then again with 10 µL of TBST solution, to remove unbound material. To determine optimal BSA concentration during the blocking step, we coated filter paper with different concentrations (1%, 0.5%, 0.1%) of blocking solution and compared reaction results to blister fluid harvested from DPTB patients and control blood serum. Mirroring the expected conventional plate ELISA results, our data showed that 1% of BSA displayed the best blocking effect for burn blister fluid when compared to our healthy human blood serum control (Appendix A). In addition to optimizing blocking, determining the best washing volume is essential for removing unbound reagents and reducing background signal. Unlike the conventional plate ELISA wash volume of 300 µL, P-ELISA requires only minute wash volumes. We tested the efficacy of three different wash solution amounts, 15 µL, 10 µL, and 5 µL, and found that three separate washes (the typical number of washes for conventional ELISA) with 5 µL removed all unbound nonspecific material (Appendix A). While the typical number of washes with conventional ELISA is three, we discovered that two cycles, one with 5 µL of wash buffer and one with 10 µL of wash buffer, provided the lowest background and strongest signal strength for P-ELISA (Appendix A).

Paper-based diagnostics provide a low-cost and easy-to-handle approach for studying a variety of target factors. Here, we demonstrated this approach by adding burn blister fluids and select reagents to test zones to detect angiogenin levels. As shown in Figure 1, burn fluid was added by hand, reagent was spotted onto each test zone, and the reaction was allowed to proceed for several minutes. Primary and secondary antibody were then added and optimized washing processes carried out to remove unbound antibody from the paper. Colorimetric responses were digitally recorded before and after color development. By analyzing the change in grayscale intensity of before and after images, we could determine the amount of angiogenin in burn blister fluids.

Relative mean intensity for angiogenin from SPTB burn blister fluids (Figure 2a) and DPTB burn blister fluids (Figure 2b) were tested and compared. In keeping with our previous study results, we found that angiogenin levels were significantly higher in DPTB fluids (4.2 ± 1.5, 95% confidence interval, 1.3–6.5, *n* = 6) than SPTB fluids (1.4 ± 0.3, 95% confidence interval, 0–4.0, *n* = 6) and healthy blood serum samples (0.9 ± 0.3, 95% confidence interval, 0.4–1.2, N = 7, *p* < 0.01) as measured by P-ELISA (Figure 2c). This demonstrates the reliability of P-ELISA to determine burn severity by evaluating angiogenin levels in burn blister fluids. Burn fluid samples have previously been tested via conventional ELISA plates for angiogenin and VEGF. Angiogenin is a downstream molecule of VEGF-regulated angiogenesis [23]. Observation of the role of VEGF in burn wound determination is interesting. Although no significant differences in VEGF were observed (SPTB: 31.5 ± 3.6 ng/mL, 95% confidence interval, 20.0–42.9 ng/mL, DPTB: 47.1 ± 10.9 ng/mL, 95% confidence interval, 16.8–77.5 ng/mL), there was a trend for higher angiogenin concentration in DPTB fluids compared with SPTB fluids as determined by conventional plate ELISA analysis (SPTB: 119.2 ± 28.4 ng/mL, 95% confidence interval, 28.8–209.6 ng/mL, DPTB: 331.5 ± 81.0 ng/mL, 95% confidence interval, 123.3–539.7 ng/mL, Figure 3, Appendix A). No significant differences in VEGF level were observed between two different burn fluids, as shown in Figure 3, which was consistent with our previous study and another study indicating that VEGF was not responsible for differentiation of circulating angiogenic cells [9] or tumor growth and angiogenesis [23]. The consistent comparability of our P-ELISA results with conventional plate ELISA results was further supported by a correlation test using Pearson’s correlation analysis. A moderate positive correlation (rho = 0.5906, *p* = 0.0722) between the titer of the plate ELISA and the relative intensity of P-ELISA was observed and is displayed in Figure 4.

In our laboratory, we have developed and explored the capacities for P-ELISA made of simple filter paper. To support the concept of P-ELISA material and process viability, we used patterned filter paper to collect directly absorbed blister fluid from human burn wounds (Appendix A) and had the test paper delivered to our laboratory on dry ice within 24 h, where we used it to successfully detect angiogenin levels via P-ELISA (Figure 5).

## 4. Discussion

Advances in diagnostic techniques have allowed clinicians to monitor disease severity in a rapid and noninvasive fashion. With regard to diagnosis of patients with ocular disease, P-ELISA provided a fast and sensitive VEGF assay of aqueous humor to monitor diseases such as senile cataracts, proliferative diabetic retinopathy, age-related macular degeneration, and retinal vein occlusion [17]. P-ELISA has also demonstrated the capacity for complex material collection capacity and multivalue measurement. It has, for instance, been used for outer macromolecule elimination and inner cervicovaginal fluid absorption to detect lactate concentration, glycogen concentration, and pH value in female genital diseases [19]. For patients with bullous pemphigoid, P-ELISA also delivered a simpler and faster diagnostic tool for detection of noncollagenous 16A (NC16A). By examining NC16A concentration with P-ELISA, bullous pemphigoid presence and disease state can be easily identified [20].

Previous concepts for burn depth assessment depend on measurements of tissue perfusion [24]. Laser Doppler imaging (LDI) studies are one of the most popular clinical techniques to assess burn depth. LDI is advantageous due to high accuracy and reduced invasiveness [25,26,27]. However, most clinical experiences using these modalities do not include examinations with intact blisters, which is a confounding factor that may skew analyzed results.

In this study, we sought to determine the burn depth by measuring angiogenin levels in burn blister fluid using the P-ELISA method. Evaluation of burn fluid contents by conventional ELISA has been successfully used to measure burn depth [9]. However, the necessity for longer operation duration severely hampered plate ELISA techniques for clinical practice. P-ELISA features operation process advantages compared to conventional plate ELISA. This P-ELISA technique required only 15 μL of reagent and an hour of processing compared to the 550 μL and 7 h–8 h required for conventional plate ELISA. Additionally, conventional plate ELISA requires a plate reader and P-ELISA results can be recorded with a camera (Table 1). Altogether, P-ELISA offers clear advantages for rapid and minimally invasive burn depth diagnosis and wound management. It may provide an easy and cost-effective method for any healthcare provider to assess burn wounds with excellent diagnostic precision and without the obstacle of a learning curve. However, P-ELISA is still conducted in a qualitative manner till now. Further refinement is needed to improve our device.

Wound pH may affect the healing process. A lower pH value was reported to be favorable for wound healing [28]. One study showed that healing burn wounds displayed a lower pH level (7.32) compared to unhealing burn wounds (pH 7.73) [29]. We observed an elevated alkaline pH value (8–9) in our burn fluid samples (Appendix A). Aspiration of burn fluids in the early stages of injury may be responsible for this phenomenon. This finding is also consistent with a previous study showing that initial stages of healing generated a more basic pH compared to the relatively acidic environment of a repaired wound [30].

We feel that P-ELISA shows great promise for potential use in clinical practice. Although the clinical application of P-ELISA in burn wound assessment remains to be further studied, we look forward to seeing the application of this new technique in burn wounds. More cases are needed to verify our hypothesis prior to clinical adoption. Further optimization of the paper pattern and additional clinical trials will improve and advance the process and its implementation as a possible tool for healthcare.

## 5. Conclusions

We designed a less-invasive and faster method to assess burn depth using P-ELISA to determine angiogenin levels from burn blister fluid. This approach is cost-effective, easy to use, and potentially accurate. Despite a moderate correlation between conventional plate ELISA and P-ELISA, we demonstrated that P-ELISA is feasible to quantify blister angiogenin levels with distinct results to diagnose SPTB and DPTB wounds. We expect this novel medicine study to pave a possible path toward a simple diagnostic method for assessing burn depth without difficulty.

## Figures and Tables

**Figure 1 diagnostics-10-00127-f001:**
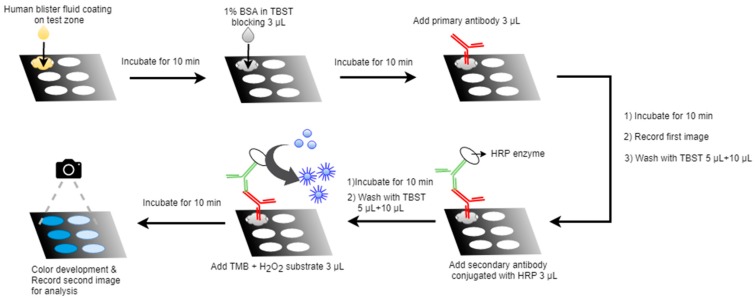
Procedure of paper-based ELISA (P-ELISA) for angiogenin analysis in burn blister fluid. Three microliters of burn fluid was loaded onto the test zone, and the test paper was incubated for 10 min. Three microliters of 1% BSA was then added for blocking, and the test paper was incubated for 10 min. The first image was recorded after loading 3 μL primary antibody, and the test paper was incubated for 10 min. The test paper was washed with 5 μL and then 10 μL TBST (Tris-buffered saline + 0.05% Tween 20) before adding 3 μL of horseradish peroxidase (HRP)-conjugated secondary antibody. The final procedure was to wash again with TBST and add 3 μL of substrate solution. After 10 min of incubation, we recorded the second image.

**Figure 2 diagnostics-10-00127-f002:**
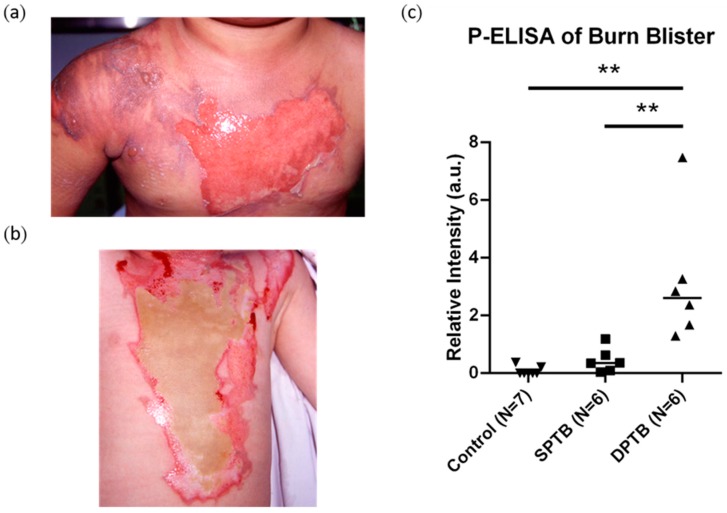
Analysis of angiogenin levels from superficial partial-thickness burn (SPTB) and deep partial-thickness burn (DPTB) blister fluids using P-ELISA. (**a**,**b**) Clinical pictures of superficial partial-thickness burn (SPTB, **a**) and deep partial-thickness burn (DPTB, **b**) wounds. (**c**) Comparison of angiogenin levels from two different burn fluids and healthy human blood serum as the control (*n* = 6 in SPTB and DPTB, *n* = 7 in control, mean ± S.D, ** *p* < 0.01).

**Figure 3 diagnostics-10-00127-f003:**
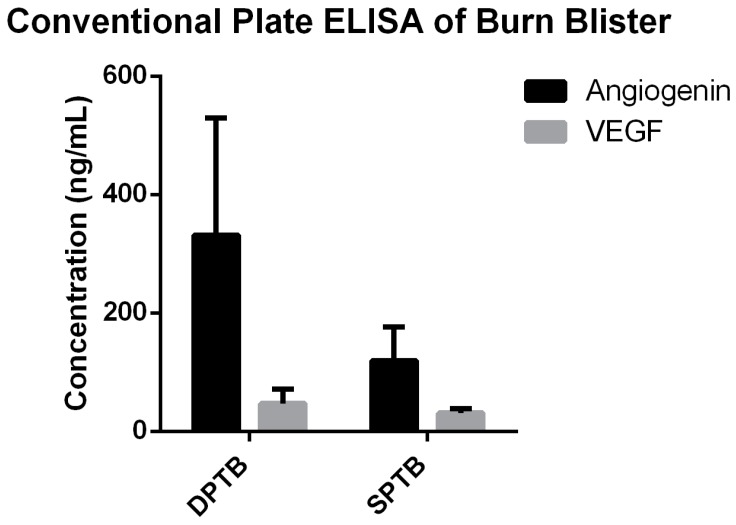
Analysis of angiogenin and VEGF (Vascular endothelial growth factor) concentrations in SPTB and DPTB blister fluids with conventional plate ELISA. A trend toward higher angiogenin concentration in DPTB fluids was detected, compared to SPTB (*n* = 4 in SPTB, *n* = 6 in DPTB, mean ± S.D.; *p* = 0.07). No significant difference in VEGF levels was observed between two different blister fluids (*n* = 4 in SPTB, *n* = 5 in DPTB, mean ± S.D.; *p* = 0.26).

**Figure 4 diagnostics-10-00127-f004:**
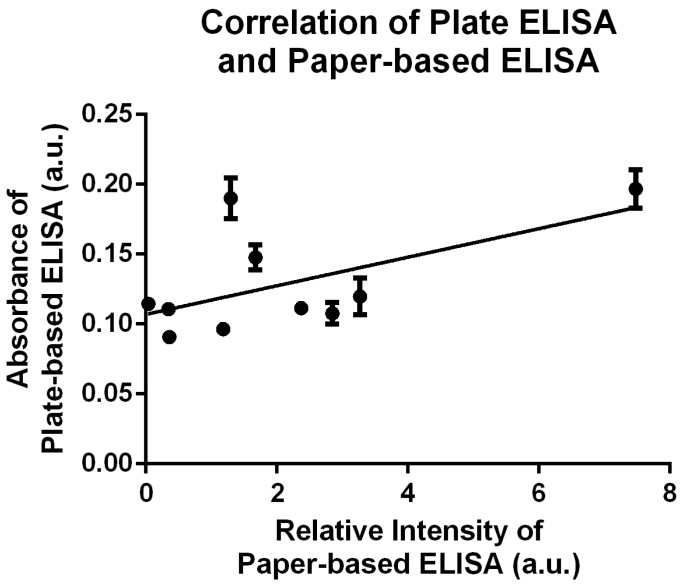
Correlation of the angiogenin detection between conventional plate ELISA and paper-based ELISA in burn blister fluids. The data show a moderate correlation between the results of P-ELISA and conventional plate ELISA (*r* = 0.5906, *p* = 0.0722).

**Figure 5 diagnostics-10-00127-f005:**
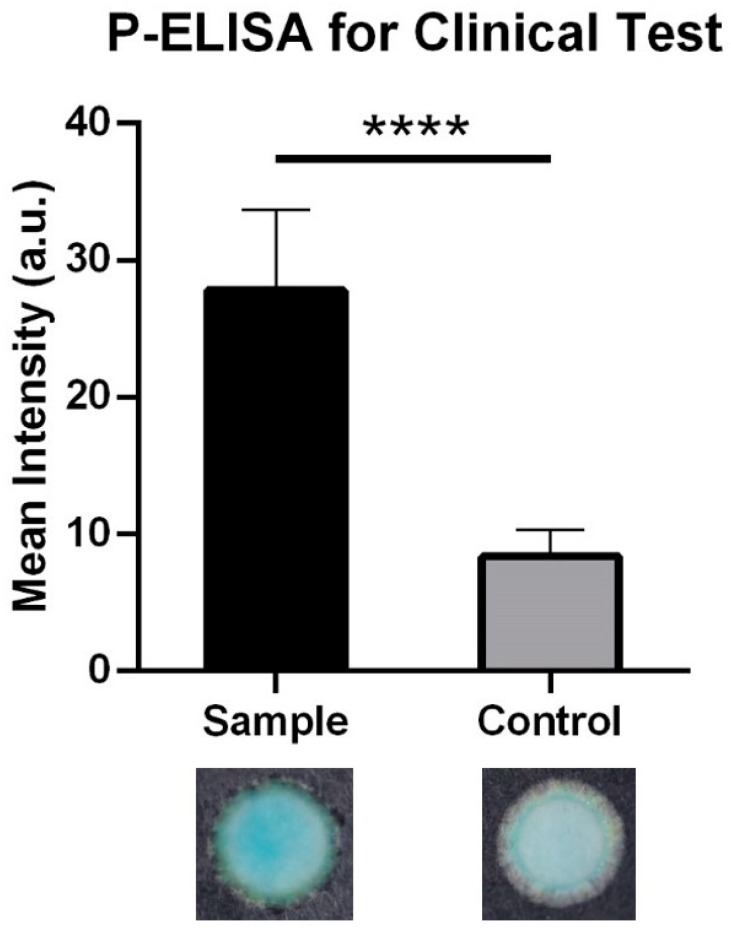
Clinical examination of angiogenin concentration by paper-based ELISA. Test paper was designed to absorb blister fluid from burn patients. Angiogenin signals were captured and analyzed. The mean intensity of detected angiogenin in burn blister fluid was significantly higher than that in normal serum control (mean ± S.D.; **** *p* < 0.0001, *n* = 6).

**Table 1 diagnostics-10-00127-t001:** Comparison of paper-based ELISA and conventional plate ELISA.

	Paper-Based ELISA		Conventional Plate ELISA	
antigen	angiogenin		angiogenin	
Primary antibody	Rabbit polyclonal anti-human angiogenin		Mouse monoclonal anti-human angiogenin	
Secondary antibody	Goat anti-rabbit lgG (HRP)		Goat anti-mouse lgG (HRP)	
Enzyme/substrate	HRP/TMB + H_2_O_2_		HRP/TMB + H_2_O_2_	
Detection device	camera		plate reader	
analysis	qualitative		quantitative	
Time and reagents	Volume (μL)	Time (min)	Volume (μL)	Time (min)
(1) antigen immobilization	3	10	50	120
(2) blocking	3	10	200	120
(3) primary antibody	3	10	100	120
(4) secondary antibody	3	10	100	120
(5) signal amplification	3	10	100	15
total	15	50	550	495

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
