# Peer review of "Preliminary Assessment of Burn Depth by Paper-Based ELISA for the Detection of Angiogenin in Burn Blister Fluid—A Proof of Concept"

_diagnostics, 2020, doi:10.3390/diagnostics10030127_

Round 1

Reviewer 1 Report

Well written study generally useful. 
I need the authors to describe how they removed the blister fluid. 
Authors must explain why they did not prefer single tests over multiple tests as patients rarely come in big burn numbers

we would appreciate the authors pointing to any innovation in the technology as I did not see any. Extra innovation section if there is any. 

otherwise a well appreciated study

Author Response

Our point to point responses to the comments:

Response to Reviewer #1:

  1. I need the authors to describe how they removed the blister fluid. Authors must explain why they did not prefer single tests over multiple tests as patients rarely come in big burn numbers

Response: Burn blister fluids were blindly aspirated with needle from individually intact blisters within the first 3 days following injury. The blister skin was not removed until the wound has been healed. We would like to revise the text in page 2, line 80.

  1. We would appreciate the authors pointing to any innovation in the technology as I did not see any. Extra innovation section if there is any.

Response: In this article, we demonstrate the feasibility of using paper-based detection method to assess the burn depth. To our knowledge, this is the first paper to demonstrate the technique of paper-based ELISA in burn wound assessment. We added this part in page 2, line 71.

Reviewer 2 Report

P-ELISA has drawn great attention because it has some obvious advantages over the previously developed ELISA such as easy operation, low cost, simple instrument and suitability for the point of care testing. In this manuscript, P-ELISA was used to assess burn depth for the detection of angiogenin from burn blister fluid; this work is interesting to warrant its publication on Diagnostics. However, there are a few points that could be modified to further improve the quality of the manuscript.

Below are points for the authors to consider for the revision:

(1) In the optimization part (Figure S1 to Figure S5), the authors should use quantitative analysis instead of qualitative analysis. For example, in line 134, the authors stated that “we found that 0.33μg/mL demonstrated the highest signal and lowest background for our primary antibody” but I could not find the highest signal when 0.033 μg/mL secondary antibody was used in the Figure S2. I suggest the authors could add the information of the signal-to-background (S/B) ratio in the optimization part of all Figures and try to revise this part more clearly.

(2) Following the comment 1, were all sample 2 (from Figure S1 to S5) from the same source? Why did the signal change so much for the washing conditions? For example, in Figure S4, the intensities of the sample 2 were all less than 5 a.u.; however, in Figure S5, all intensities of the sample 2 were higher than 15 a.u. Please explain this part.

(3) For the detection procedure (Figure 1), the first three steps do not seem to need wash with TBST. Does it not affect the detection performance such as sensitivity or specificity of the P-ELISA?

(4) A polyclonal antibody has higher potential for cross reactivity which affects the diagnostic efficacy. Thus, in Table 1, why did the authors use a polyclonal antibody as a primary antibody for angiogenin?

Author Response

Our point to point responses to the comments:

Response to Reviewer #2:

  1. In the optimization part (Figure S1 to Figure S5), the authors should use quantitative analysis instead of qualitative analysis. For example, in line 134, the authors stated that “we found that 0.33μg/mL demonstrated the highest signal and lowest background for our primary antibody” but I could not find the highest signal when 0.033 μg/mL secondary antibody was used in the Figure S2. I suggest the authors could add the information of the signal-to-background (S/B) ratio in the optimization part of all Figures and try to revise this part more clearly.

Response: Thank you for your suggestion. We have added the information of the S/B ration in each optimization Figures. Furthermore, in Line 135, the “color intensity” is changed into “signal”, and in Line 138, we revised the sentence into “we found that 0.33µg/mL demonstrated the highest signal-to-background ratio for our primary antibody ”

  1. Following the comment 1, were all sample 2 (from Figure S1 to S5) from the same source? Why did the signal change so much for the washing conditions? For example, in Figure S4, the intensities of the sample 2 were all less than 5 a.u.; however, in Figure S5, all intensities of the sample 2 were higher than 15 a.u. Please explain this part.

Response: Thank you for your question. All sample 2 comes from the same source, the reason that the mean intensity of sample 2 in Figure S3 and Figure S4 is low is because of some complex in the blister sample. Therefore, in Figure S3 and S4, we took the result of sample 1 and 3 as our main consideration during optimization process.

  1. For the detection procedure (Figure 1), the first three steps do not seem to need wash with TBST. Does it not affect the detection performance such as sensitivity or specificity of the P-ELISA?

Response: The first three steps include adding sample, blocking reagent, and then primary antibody. These reagents can react mix together in the test zone since some ELISA kits also combine the blocking reagent with detection antibody solution and add immediately after the sample adding process. This is because that even though excess regent is added during the first three process, it can be removed at the fourth step of washing without interfering the result.

  1. A polyclonal antibody has higher potential for cross reactivity which affects the diagnostic efficacy. Thus, in Table 1, why did the authors use a polyclonal antibody as a primary antibody for angiogenin?

Response: We used polyclonal antibody because the angiogenin level rather high (nanogram per milliliter level), and also we tried to develop qualitative detection rather then quantitative detection. Therefore, we selected polyclonal for its wider range of target (faster reaction) and on the other hand, modified the blocking concentration to reduce the cross reactivity.

Reviewer 3 Report

The authors report on a paper-based ELISA for burn depth classification based on angiogenin level. Although this approach is promising, the points below should be addressed prior to publication:

Figure S1-S6 should include error bars. Recent literature should be cited and discussed in the introduction (ex., https://doi.org/10.3389/fbioe.2019.00069; https://doi.org/10.1021/acsnano.6b05610)

A receiver operating characteristic curve (ROC curve) should be performed and discussed.

Clinical specificity and selectivity of the studied device should be determined and discussed.

Washing procedures in the resulting P-ELISA are not clear. How washed reagents are removed from the paper-based sensing surface? Generally, ELISA washing procedures involve addition and aspiration of a washing buffer, thereby removing washed reagents from the sensing surface; however, it is not clear if such a washing procedure requires aspiration or a drying process in P-ELISA. Given the wettability/absorption limit of the utilized paper this seems difficult to achieve in the utilized substrate. Please elaborate.

At least 60 control samples, 30 samples of SPTB and 30 samples of DPTB should be analyzed to show a statistically acceptable model of the studied device.

Figure 4 should include error bars.

Limit of detection, dynamic range, accuracy and precision should be included in Table 1.

Author Response

Our point to point responses to the comments:

Response to Reviewer #3:

  1. Figure S1-S6 should include error bars. Recent literature should be cited and discussed in the introduction (ex., https://doi.org/10.3389/fbioe.2019.00069; https://doi.org/10.1021/acsnano.6b05610)

Response: Thank you for the comment. It is not sufficient to make error bars because optimization process is conducted only once or twice time. Different kinds of conditions are compared and selected in Figures S1 to S6. However, it may not be the best for P-ELISA detection of angiogenin in burn blister fluid. We have cited the papers in page 2, line 76.

  1. A receiver operating characteristic curve (ROC curve) should be performed and discussed. Clinical specificity and selectivity of the studied device should be determined and discussed.

Response: We agree with your points. However, this study was focused on proof of concept. More studies and samples are needed to determine the optimal cut-off point.

  1. Washing procedures in the resulting P-ELISA are not clear. How washed reagents are removed from the paper-based sensing surface? Generally, ELISA washing procedures involve addition and aspiration of a washing buffer, thereby removing washed reagents from the sensing surface; however, it is not clear if such a washing procedure requires aspiration or a drying process in P-ELISA. Given the wettability/absorption limit of the utilized paper this seems difficult to achieve in the utilized substrate. Please elaborate.

Response: Thank you for the comment. The washing process requires a blotting paper at the bottom of the sensing paper. Once we drop the washing buffer on the test zone, washing buffer penetrate the sensing paper and absorbed by the blotting below under capillary force. We added the description in page 3, line 106.

  1. At least 60 control samples, 30 samples of SPTB and 30 samples of DPTB should be analyzed to show a statistically acceptable model of the studied device.

Response: Thanks for your suggestion. As mentioned above, it is a proof-of-concept study. Although clinical applications of P-ELISA for burn wound assessment requires additional study, we look forward to seeing further growth in the field.

  1. Figure 4 should include error bars.

Response: Thank you for the suggestion. We have added the error bars in page 6, Figure 4.

  1. Limit of detection, dynamic range, accuracy and precision should be included in Table 1.

Response: Thank you for the suggestion. In this study, paper-based ELISA method can only show different cytokine expressions between deep and superficial partial thickness burn wounds. Further study should be done to estimate the LOD and LOQ value as well as the accuracy and precision of the detection. Although it can only conduct qualitative analysis in recent stage, P-ELISA exhibits its advantage in wound assessment when compared with conventional ELISA in Table 1. We will add the analytic method in page 8, Table 1, and revise the text in page 8, line 244.

Round 2

Reviewer 2 Report

I am very happy that revised manuscript has been improved significantly in response to the reviewer comments. Only one point should be corrected: in line 138, the optimal concentration is 0.033 μg/mL, not 0.33 μg/mL. Please check and revise it.

Author Response

Response to Reviewer #2:

  1. In line 138, the optimal concentration is 0.033 μg/mL, not 0.33 μg/mL. Please check and revise it.

Response: Thank you for your reminder. We had revised the concentration value in page 4, line 138.

Reviewer 3 Report

The auhtors partially addressed the comments and suggestions (previously provided by the referee) discussing that this is a proof of concept. Hence, the title of this manuscript is suggeted to be changed as follows: "Preliminary Assessment of Burn Depth by Paper-based ELISA for the Detection of Angiogenin in Burn Blister Fluid -A proof of concept".

Author Response

Response to Reviewer #3:

  1. The authors partially addressed the comments and suggestions (previously provided by the referee) discussing that this is a proof of concept. Hence, the title of this manuscript is suggested to be changed as follows: "Preliminary Assessment of Burn Depth by Paper-based ELISA for the Detection of Angiogenin in Burn Blister Fluid -A proof of concept".

Response: Thank you for your suggestion. We agree your point and had changed the title to "Preliminary Assessment of Burn Depth by Paper-based ELISA for the Detection of Angiogenin in Burn Blister Fluid -A proof of concept".